Article 

# T cell-independent eradication of experimental glioma by intravenous TLR7/8-agonist-loaded nanoparticles

Verena Turco[1,2,3], Kira Pfleiderer[1,3], Jessica Hunger [1,3,4], Natalie K. Horvat[4,5,6], Kianush Karimian-Jazi [3], Katharina Schregel[3], Manuel Fischer[3], Gianluca Brugnara [3], Kristine Jähne[1,2], Volker Sturm [3], Yannik Streibel[3], Duy Nguyen[7], Sandro Altamura [5,6], Dennis A. Agardy [1,2,4], Shreya S. Soni[8], Abdulrahman Alsasa[8], Theresa Bunse[1,2], Matthias Schlesner [7,9], Martina U. Muckenthaler [5,6], Ralph Weissleder [10,11], Wolfgang Wick [12,13], Sabine Heiland[3], Philipp Vollmuth [3], Martin Bendszus[3], Christopher B. Rodell[8], Michael O. Breckwoldt [1,3,14] ✉ & Michael Platten [1,2,14] ✉

Glioblastoma, the most common and aggressive primary brain tumor type, is considered an immunologically "cold" tumor with sparse infiltration by adaptive immune cells. Immunosuppressive tumor-associated myeloid cells are drivers of tumor progression. Therefore, targeting and reprogramming intratumoral myeloid cells is an appealing therapeutic strategy. Here, we investigate a β-cyclodextrin nanoparticle (CDNP) formulation encapsulating the Toll-like receptor 7 and 8 (TLR7/8) agonist R848 (CDNP-R848) to reprogram myeloid cells in the glioma microenvironment. We show that intravenous monotherapy with CDNP-R848 induces regression of established syngeneic experimental glioma, resulting in increased survival rates compared with unloaded CDNP controls. Mechanistically, CDNP-R848 treatment reshapes the immunosuppressive tumor microenvironment and orchestrates tumor clearing by pro-inflammatory tumor-associated myeloid cells, independently of T cells and NK cells. Using serial magnetic resonance imaging, we identify a radiomic signature in response to CDNP-R848 treatment and ultrasmall superparamagnetic iron oxide (USPIO) imaging reveals that immunosuppressive macrophage recruitment is reduced by CDNP-R848. In conclusion, CDNP-R848 induces tumor regression in experimental glioma by targeting blood-borne macrophages without requiring adaptive immunity.

Glioblastoma, the most common and aggressive primary brain tumor in adults, is considered an immunologically "cold" lymphodepleted tumor, due to its growth in an immune-privileged site, low mutational load, and a highly immunosuppressive tumor microenvironment (TME)[1]. Tumor-associated myeloid cells (TAMs) constitute a significant part of the tumor mass and orchestrate tumor progression and resistance to immunotherapies such as immune checkpoint blockade (ICB). During tumor progression, monocyte-derived macrophages are recruited to the TME as crucial drivers of tumor-associated immunosuppression[2].

Toll-like receptors (TLR) play an essential role in initiating innate immunity by recognizing pathogen-associated molecular patterns

(PAMPs) and are well-established drivers for viral and bacterial sensing as well as immune responses[3]. TLR signaling has been described as an essential pathway facilitating inflammatory programs in glioma-associated myeloid cells[4]. Many imidazoquinolines are synthetic TLR activators, including resiquimod (R848), a potent dual TLR7/8 agonist that has recently gained attention as a cancer immunotherapy agent due to its ability to reprogram TAMs in subcutaneous tumor models[5–7]. Systemic administration of TLR agonists has been limited, due to side effects reflective of a systemic interferon response[5], fueling the development of local delivery systems and nanotherapeutic drug carriers to enhance targeted delivery and reducing systemic drug effects[6–8].

Cyclodextrin nanoparticles (CDNP) have recently been developed to improve drug targeting to TAMs and were established for endosomal targets like the TLR7 receptor[3]. Nanoparticles encapsulating R848 (CDNP-R848) have been shown to reprogram the myeloid compartment in solid, subcutaneous tumor models towards a pro-inflammatory phenotype[9]. We hypothesized that systemic administration of nanoparticles encapsulating an immunostimulatory compound might be particularly effective in glioma, as TAMs constitute up to 50% of the tumor mass and are the main drivers of the immunosuppressive TME. Current concepts propose the recruitment of immunosuppressive blood-borne macrophages to the glioma microenvironment during tumor progression[2]. Here, we comprehensively assess the effects of systemic administration of CDNP-R848 in established experimental gliomas and find that intravenous monotherapy with CDNP-R848 leads to glioma regression and increased survival compared with CDNP vehicle. Mechanistically, CDNP-R848 treatment reshapes the immunosuppressive tumor microenvironment to a pro-inflammatory state which we comprehensively assess by myeloid-targeted imaging and immune-profiling. Thus, we show that targeting the myeloid compartment by the TLR 7/8 axis is a promising therapeutic target in glioma.

## Results

### TLR7 is highly expressed on macrophages and could serve as a therapeutic target in glioma

First, we validated TLR7/8 expression in the mouse and human glioma TME using a recently published single-cell RNA sequencing dataset of infiltrating CD45[+] immune cells in treatment-naive murine syngeneic Gl261 tumors and human glioma[10]. (Supplementary Fig. 1a–f). Notably, *TLR7* is strongly expressed on immune cells (16.8% and 18% in murine and human cells, respectively), while there is only minimal *TLR8* expression (Supplementary Fig. 1b, e). To address whether there is constitutive TLR7 activation, we further investigated the expression of TLR7 regulator triggering receptor expressed on myeloid cells like 4 (*TREML4*)[11] and downstream targets[12,13]. We found very few transcripts of *TREML4* and *interleukin (IL)12*, while genes encoding other cytokines such as *IL1* and *TNFα* displayed robust baseline expression, likely due to other activated pro-inflammatory pathways (Supplementary Fig. 1c, f). This indicates that TLR7 has minimal baseline activity and could potentially be targeted in glioma immunotherapy to induce immunostimulatory TLR7-mediated pathways.

### CDNP-R848 leads to a pro-inflammatory shift and antitumor activity of macrophages in vitro

We next examined the effect of CDNP-R848 on the inflammatory phenotype of bone marrow-derived macrophages (BMDMs) in vitro. CDNP-R848 induced a strong pro-inflammatory phenotype with increased mRNA levels of *nitric oxide synthase 2 (Nos2)* and *lipocalin 2 (Lcn2)*, as well as protein upregulation of CD80, CD64, and MHCII, whereas anti-inflammatory markers such as CD71, CD172 and Mer tyrosine kinase (MerTK) were downregulated (Fig. 1a, b). By contrast, the pro-inflammatory cytokines *IL6*, *IL1ß*, and *IL12* were upregulated (Fig. 1a). To functionally characterize macrophages after CDNP-R848 treatment and evaluate a potential direct antitumor effect, we investigated the production of reactive oxygen species (ROS) by macrophages after CDNP-R848 treatment in vitro. Indeed, CDNP-R848-treated BMDMs produced higher amounts of ROS (Fig. 1c). Furthermore, BMDMs co-cultured with Gl261 glioma cells and treated with CDNP-R848 induced glioma cell death, likely orchestrated by the pro-inflammatory shift of macrophages and increased ROS production (Fig. 1d). A potential direct antitumor cell effect was also excluded in vitro: although Gl261 cells do take up CDNP-R848 in vitro (Supplementary Fig. 2a), this did not affect tumor cell proliferation (Supplementary Fig. 2b, c) nor induce the production of cytokines or chemokines (Supplementary Fig. 2d).

### Macrophages and microglia are efficiently targeted by CDNP-R848 in vivo

After in vitro validation, we assessed CDNP-R848 efficacy in the syngeneic Gl261 glioma model in vivo. For effective glioma therapy, the blood-brain barrier (BBB) is a major hurdle for delivering drugs to the brain[14]. Therefore, we assessed whether both glioma-associated macrophages and microglia can be targeted by intravenous CDNP-R848 administration using immunohistochemistry and flow cytometry. Histological analysis showed specific uptake of fluorescently labeled CDNP-R848 (CDNP-R848-VT680) at the tumor site without detectable NP incorporation by Gl261-GFP[+] tumor cells nor in the healthy adjacent brain parenchyma (Fig. 1e). Instead, we found a high proportion of monocyte-derived myeloid cells that phagocytized CDNP-R848 nanoparticles in vivo (25.7% ± 2.9% of CDNP-R848-VT680 labeled macrophages in the TME), whereas tumor microglia (4.8% ± 0.8% CDNP-R848-VT680 labeled microglia) and splenic macrophages (13.2% ± 0.7%) revealed less nanoparticle uptake ($n = 8$ mice CDNP-R848-VT680 treated mice; $n = 5$ PBS injected control mouse; two independent experiments, Fig. 1f, g; Supplementary Fig. 2e). After intravenous administration, CDNP-R848-VT680 accumulated selectively in TAMs in the glioma region and in monocyte-derived myeloid cells in the blood, lymph node and spleen. By contrast, there were no relevant levels in the liver, bone marrow kidney, lung, or the contralateral, tumor-free hemisphere (Supplementary Fig. 2e–g). Both CDNP-R848 and R848 administration was well tolerated but led to a transient weight loss of 13.1% and 17.1% compared to PBS control injected animals. However, the weight loss was fully reversible after completing the treatment cycle (Supplementary Fig. 3a, b). There were no detectable differences in hematological parameters during treatment further indicating adequate drug tolerability (Supplementary Fig. 3c).

### CDNP-R848 induces tumor regression and prolongs survival in the Gl261 glioma model

We next tested whether activation of the TLR7/8 pathway by CDNP-R848 can control the growth of established orthotopic gliomas in a syngeneic mouse glioma model. Gl261 tumors were grown for two weeks and randomized to either CDNP-R848, R848, CDNP vehicle control or PBS after baseline MRI (Fig. 2a). Response to CDNP-R848 was evaluated by longitudinal ultrahigh field MRI with previously established response criteria[15]. Tumor volume was assessed weekly (throughout weeks 2 to 4 after tumor inoculation) using T2-weighted sequences (Fig. 2b). Based on the relative change in tumor volume, treatment response was classified as complete response (CR), partial response (PR), stable disease (SD), or progressive disease (PD) utilizing modified iRANO criteria as previously established[15]. Mice showing CR, PR, and SD were defined as responder mice (R), while mice with PD were classified as non-responders (NR)[15]. Whereas Gl261 tumors grew exponentially in the CDNP vehicle or PBS-treated control groups ($n = 6$ mice, 100% progressive disease), CDNP-R848 administration resulted in 62.5% partial response, 25.0% stable disease and 12.5% progressive disease ($n = 8$ mice, (Fig. 2c, d). Importantly, CDNP-R848 was also superior to the unformulated drug R848, which resulted in stable disease in 33.3% of animals but did not achieve PR or CR (Fig. 2e, f). The robust

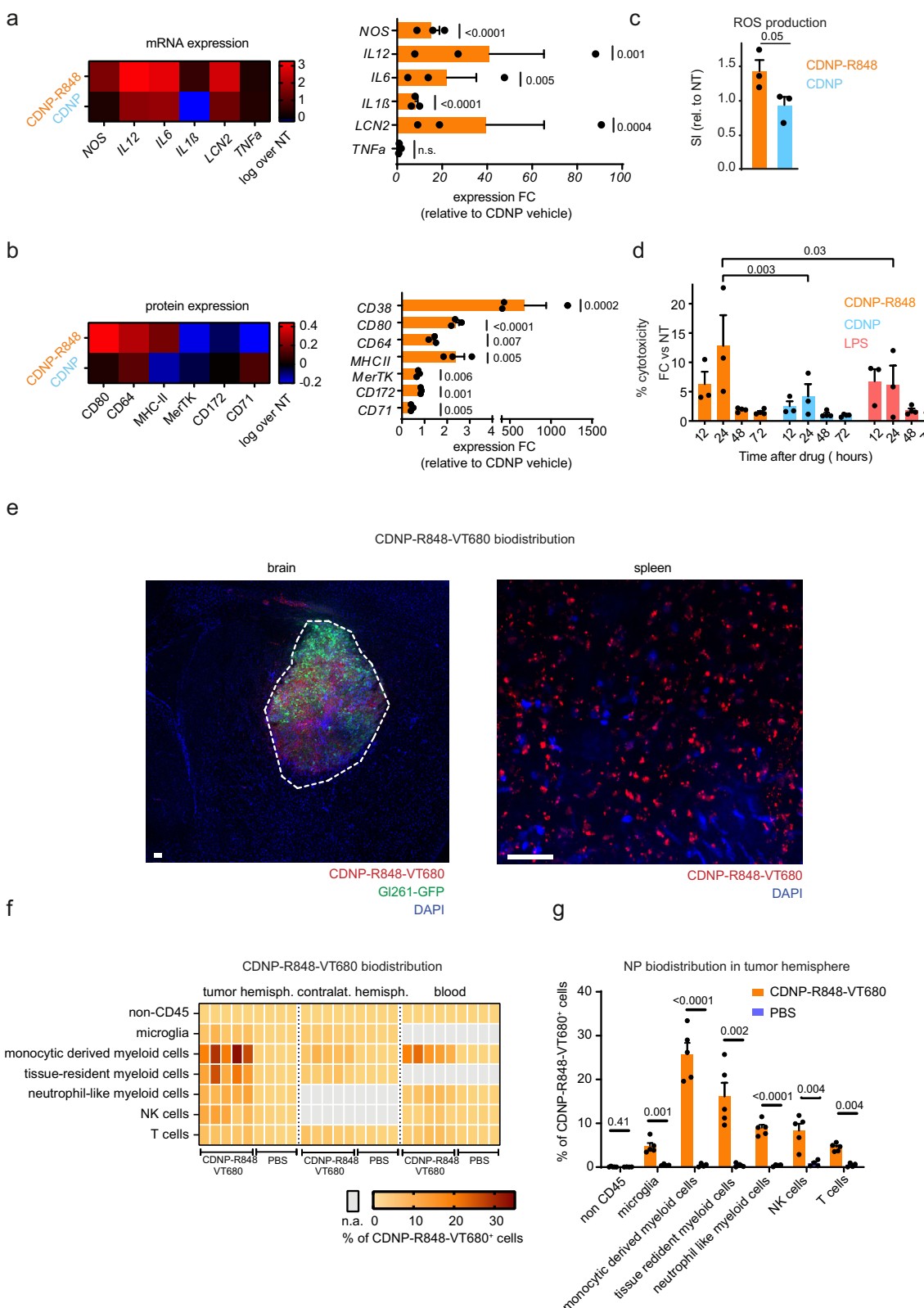

therapeutic effect of CDNP-R848 therapy was further confirmed in additional in vivo experiments that led to an overall response rate of 83.0% (CDNP-R848: CR: 4.6%; PR: 49.2%; SD: 29.2%; PD: 17.0% vs CDNP: PD: 100%; mean tumor volume in week 4: 8.6 mm³ vs 46.8 mm³; $p < 0.001$, Supplementary Fig. 3d, e). Moreover, CDNP-R848 led to a significant increase in median overall survival to 65 days compared to 28 days in CDNP vehicle-treated animals ($n = 7$ mice per group,

Supplementary Fig. 3f) and 3/7 mice (43%) in the CDNP-R848 group remained clinically intact and were long term survivors with no sign of tumor mass on three subsequent MRI measurements until day 85 after tumor cell inoculation.

Tumor monitoring by MRI showed 31.5% of responding animals ($n = 17/54$ mice) exhibited an initial tumor volume increase >40%, followed by subsequent tumor regression in week 4 (Supplementary

**Fig. 1 | CDNP-R848 selectively targets macrophages and shifts them towards a pro-inflammatory phenotype. a** mRNA expression profiling by qRT-PCR in bone marrow-derived macrophages (BMDMs) after 1 μM CDNP-R848 or CDNP vehicle treatment Expression of *Nos2*, *Il-6*, *tnfα*, *Il-1ß*, *lcn2* and *Il-12 was assessed*. (**a**–**c**: *n* = 3 BMDM cultures per group from three mice). **b** Protein expression profiling by flow cytometry in BMDMs after CDNP-R848 or CDNP vehicle treatment. Quantification of CD80, CD64 and CD38, CD71, CD172, and MerTK. Results are normalized to non-treated cells (NT) BMDMs. **c** Quantification of ROS production in BMDMs 24 h after CDNP-R848 (1 μM) or CDNP vehicle treatment as measured by flow cytometry after incubation with the probe CellROX Green. Non-treated (NT) cells served as negative control. **d** Gl261 glioma cells were co-cultured with BMDMs upon CDNP-R848, CDNP vehicle or LPS treatment. Cytotoxicity was quantified by LDH release. Values are corrected for spontaneous effector and target cell LDH release. *n* = 3 to 4 BMDM cultures per group from three to four mice per time point. **e** Representative

histological image from three biological replicates (*n* = 3 mice) shows NP accumulation in the TME of Gl261 glioma but not in the adjacent healthy brain. Dashed line indicates glioma border. Micrograph of whole-mounted spleen shows NP uptake in splenic phagocytes. Scale bar is 100 μm. **f**, **g** Biodistribution analysis of CDNP-R848-VT680 within myeloid cells 24 h after a single, intravenous dose as assessed by flow cytometry of monocyte-derived myeloid cells (CD45⁺ CD11b^high, Ly6c⁺), tissue-resident myeloid cells (CD45⁺ CD11b^high, F4/80⁺), neutrophil like myeloid cells (CD45⁺ CD11b^high, CD11c^high), NK cells (NK1.1⁺), T-cells (CD3⁺) and microglia (CD45⁺ CD11b^intermediate) in the tumor-bearing hemisphere, contralateral hemisphere and blood. *n* = 5 CDNP-R848-VT680 mice and 4 PBS-treated animals. Data are representative of two independent experiments and presented as mean ± SEM. Statistical significance was determined by two-way ANOVA with Sidak's test. n.a.: not applicable. NT: non-treated. FC: fold change.

Fig. 3g–i). This phenomenon, known as pseudoprogression (PsPD), has been observed in response to radiotherapy and specific immunotherapeutic interventions in glioma clinical trials and is generally appreciated as indicative of the intratumoral inflammatory response[16,17].

## CDNP-R848 treatment is associated with pro-inflammatory changes in the glioma TME

To further characterize the inflammatory cascade induced by CDNP-R848 and to dissect the underlying therapeutic mechanism, we profiled cytokine and chemokine levels in the tumor-bearing hemisphere during the effector (week 3) and tumor clearing phase (week 4). During the effector phase, there was a strong production of the pro-inflammatory cytokine IL-12 subunit p40 after CDNP-R848 treatment, confirming successful TLR activation in the TME[9,18] (concentration $18.2 \times 10^5$ pg/ml compared to 7995 pg/ml in the CDNP vehicle group, 22.8-fold induction, $p < 0.001$, $n = 10$ mice, Fig. 3a). Interestingly, several chemokines implicated in macrophage proliferation, such as MIP-1α (CCL3), GM-CSF, and KC[16], were also significantly upregulated (Fig. 3a, b), signifying macrophage activation, presumably in response to TLR stimulation. Moreover, pro-inflammatory interleukin levels, such as IL-2, 9, and 13, were significantly higher. IL-12p40 remained upregulated in the tumor clearing phase (week 4) when most other cytokines returned to baseline levels (Fig. 3b). Consequently, even after a single dose of CDNP-R848, IL-12p40 and multiple pro-inflammatory chemokines and cytokines such as IFN-γ and TNF-α were strongly upregulated (Supplementary Fig. 4a, b).

## CDNP-R848 treatment reprograms the phenotype of glioma-infiltrating T cells

Antitumor treatment effects of TLR 7/8 agonists in peripheral tumors have been shown to rely on cytotoxic CD8 T cells for tumor cell clearance[2,5,8]. In search of the effector mechanism of CDNP-R848 in the glioma model, we profiled the T cell compartment within the glioma TME by flow cytometry in the effector (week 3) and clearing (week 4) phase. CDNP-R848 treatment significantly enhanced CD4 T cell infiltration and increased CD4 proliferation during the effector phase, whereas CD8 T cell numbers and CD8 proliferation were unchanged (Fig. 3c, d). Additionally, CDNP-R848 treated tumors were characterized by significantly reduced frequencies of regulatory T cells, further demonstrating a pro-inflammatory shift of the TME (Fig. 3e).

In line with our previous results, the TME in the tumor-clearing phase was dominated by CD4 T cells, whereas CD8 and regulatory T cell infiltration were markedly decreased (Fig. 3f–h). Interestingly, we observed no significant difference in expression of the immune checkpoint programmed cell death protein-1 (PD-1), while CDNP-R848-treatment led to a substantial decline of lymphocyte activating 3 (Lag3) expression in both CD4 and CD8 T cells during the tumor clearing phase (Fig. 3g). Thus, these results confirm that CDNP-R848 treatment

increases the ratio of conventional CD4 T cells over regulatory T cells as well as CD8 T cells and reduce Lag3-mediated T cell exhaustion.

## CDNP-R848 treatment efficacy is independent of T cells and NK cells

Based on our previous observations, we hypothesized that T cells are crucial for mediating response to CDNP-R848. To address this hypothesis further, we selectively depleted CD4⁺ and CD8⁺ cells during CDNP-R848 treatment using depletion antibodies in conjunction with respective isotype controls (anti-CD4; anti-CD8; Fig. 4a, b). Immune cell depletion was performed after glioma inoculation and before CDNP-R848 treatment was initiated. The success of depletion was confirmed by peripheral blood analysis and flow cytometry or immunohistochemistry of the respective immune cell population in the TME. Strikingly, tumor growth remained fully controlled by CDNP-R848, even in the absence of CD4+ and CD8+ T cells and—unlike in subcutaneous tumor models[9]—tumor growth was unchanged compared to isotype controls (tumor volume week 4: CDNP-R848 + α-CD8: 3.9 mm³ vs CDNP-R848 + isotype: 3.6 mm³, n.s.; tumor volume CDNP + α-CD4 depletion: 39.03 mm³ vs CDNP-R848 + α-CD4: 1.4 mm³, $p < 0.001$, vs CDNP-R848 + isotype: 1.0 mm³, n.s., $n = 5$–7 animals per group, Fig. 4a, b, d, Supplementary Fig. 5c, e, f). In this model, CDNP-R848 treatment efficacy was therefore independent of CD4 and CD8 cells.

We next examined whether the primary antitumor activity of CDNP-R848 is dependent on NK cells, as previously suggested in solid subcutaneous tumor models[19,20]. Remarkably, depletion of NK cells did not abrogate therapeutic efficacy of CDNP-R848 (tumor volume CDNP + α-NK1.1 depletion: 29.2 mm³ vs CDNP-R848 + α-NK1.1: 1.5 mm³, $p < 0.001$, vs CDNP-R848 + isotype: 7.8 mm³, n.s. compared to CDNP-R848 + α-NK1.1, Fig. 4c, d, Supplementary Fig. 5d, g). Thus, we exclude a major role for T cells and NK cells in mediating tumor regression of CDNP-R848 therapy.

## Blood-borne MDSC recruitment is reduced by CDNP-R848 treatment in vivo

To assess the contribution of macrophages to the therapeutic effect in vivo, we further profiled the innate immune cell compartment in the Gl261 glioma model by flow cytometry: macrophage infiltration was reduced both at the end of the treatment cycle (effector phase, week 3 after tumor inoculation) and the late disease stage (clearing phase, week 4) in the CDNP-R848 group compared to CDNP vehicle-treated mice (Fig. 5a, d, Supplementary Fig. 6a). Additional phenotyping of macrophages (CD45^high/CD11b⁺ population) showed that CDNP-R848 treatment leads to a pro-inflammatory shift of TAMs with the enhanced presence of pro-inflammatory F4/80⁺/MHCII⁺ macrophages during the effector and tumor clearing phases (Fig. 5b, e). Furthermore, CDNP-R848 treatment significantly decreased frequencies of anti-inflammatory Ly6C⁺/PD-L1⁺ myeloid-derived suppressor cells (MDSC) between tumor effector and clearing phases (Fig. 5c, f). Hence, therapeutic efficacy seems to be orchestrated solely by myeloid cells.

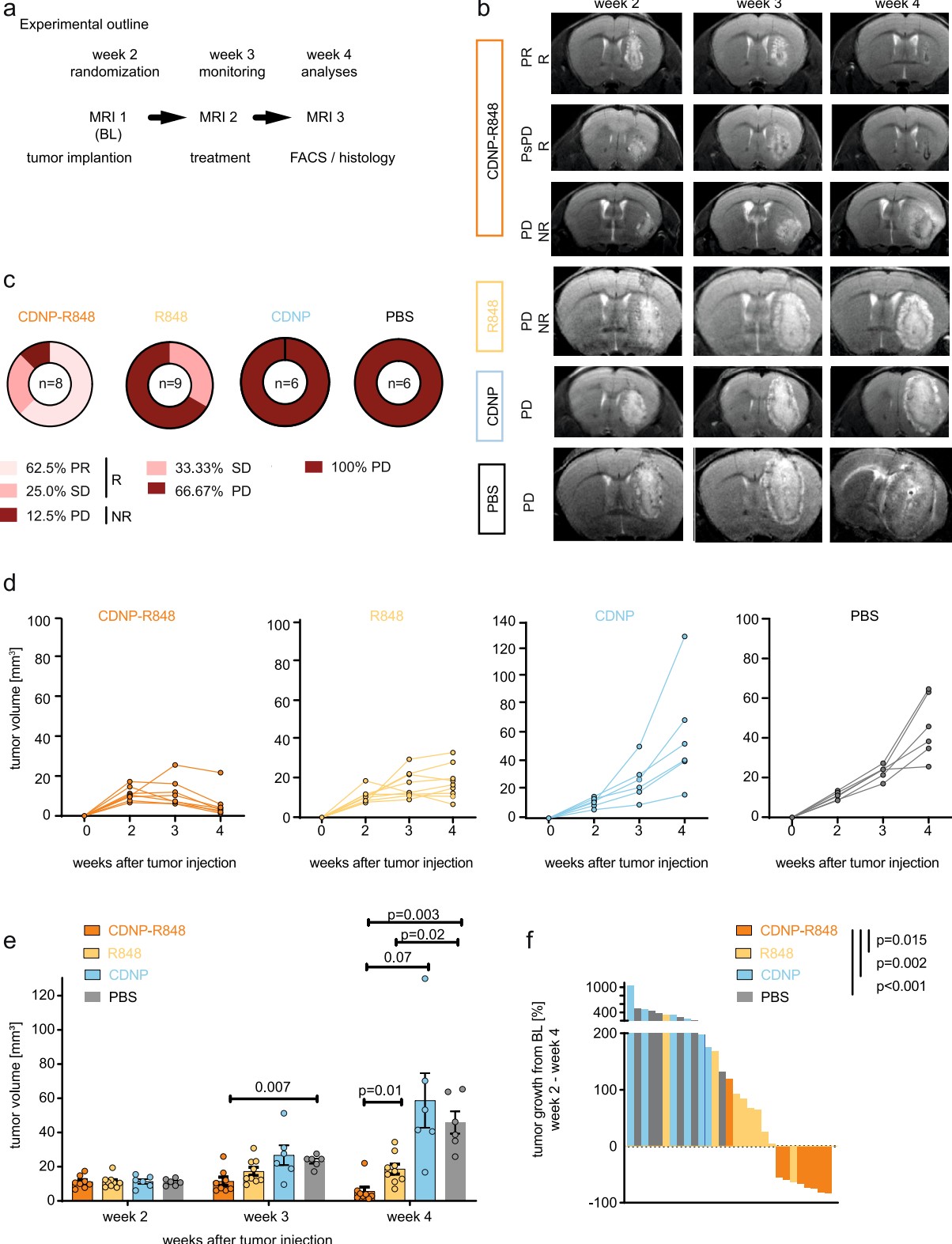

## Monitoring macrophage recruitment to the TME by USPIO imaging and imaging biomarkers of response and resistance

Given the heterogeneous response and the relevant amount of PsPD (Fig. 2c), we performed multiparametric MRI including diffusion, Gd-contrast-enhanced, and USPIO nanoparticle imaging for macrophage tracking to derive an imaging signature of response, resistance, and pseudoprogression. Whereas blood-brain barrier disruption (BBB-D), assessed by Gd-uptake, was unchanged at the end of the treatment cycle (week 3), BBB-D was lower in the clearing phase in CDNP-R848 animals (week 4, Supplementary Fig. 6b–d). The degree of BBB-D was, however, strongly correlated with tumor volume ($R^2 = 0.43$; $p = 0.003$, Supplementary Fig. 6d), indicating that it is no suitable marker of the specific underlying immunological processes. In search for a predictive marker of response and resistance, we further assessed diffusion-

**Fig. 2 | CDNP-R848 therapy is an effective monotherapy and decreases Gl261 glioma growth. a** C57Bl/6J mice were treated intravenously with 100 μl CDNP-R848, R848, vehicle control CDNP or PBS on d14, d17 and d20 after intracranial Gl261 tumor injection (d0). Mice were randomized according to tumor volume on d13 (MRI1, week 2). Tumor growth was assessed after the completion of treatment cycle on d20 (MRI2, week 3) and d26 (MRI3, week 4) **b** Representative T2-w MR images of CDNP-R848, R848 and control CDNP and PBS-treated mice. R, response; NR, non-response; PsPD, pseudoprogression; PD, progressive disease. **c** Overall response rate in CDNP-R848 treated group vs R848, CDNP vehicle-treated and PBS groups (CDNP-R848: $n = 8$ mice; R848: $n = 9$ mice; CDNP: $n = 6$; PBS: $n = 6$ mice).

Response rates were calculated based on Aslan et al.[15]. Pseudoprogression was defined as an increase of tumor volume > 40% from MRI 1 to 2 and a subsequent decrease in tumor volume to MRI 3. **d** Tumor growth curves of head to head comparison of CDNP-R848 ($n = 8$ mice), R848 ($n = 9$ mice), CDNP ($n = 6$ mice) and PBS ($n = 6$ mice). **e, f** tumor volumes and waterfall plots for CDNP-R848 ($n = 8$ mice) vs R848 ($n = 9$ mice), CDNP and PBS-treated mice ($n = 6$ mice per group, respectively). Response was assessed by % of tumor growth between from baseline (day 13) to day 26 post tumor inoculation. The data are from one independent experiment and presented as mean ± SEM. Statistical significance was determined by unpaired two-tailed Student's $t$ tests or by two-way ANOVA with Tukey's test.

weighted imaging, which has recently been described as a predictive marker for monitoring checkpoint inhibition in glioma patients[21]. Fractional anisotropy (FA) as an MRI measure of hypercellularity was unchanged in CDNP-R848 treated gliomas compared to vehicle controls (Supplementary Fig. 6e, f). To directly visualize TAMs in the TME during the effector phase in week 3, we performed USPIO imaging, an approach we have previously established in gliomas and experimental neuroinflammation for tracking macrophage recruitment upon specific uptake of USPIOs by myeloid cells[22,23] (Fig. 5g–i). Quantitative magnetic gradient echo (MGE) images were acquired before and 24 h after intravenous USPIO administration (30 mg/kg ferumoxytol). Interestingly, we found increased levels of USPIO uptake in vehicle-treated animals compared to CDNP-R848 early responding mice (Δ T2*: −11.7 ± 4.2 vs −4.0 ms ± 2.8, $p = 0.01$, $n = 9$ mice), indicating an increased influx of myeloid cells into the TME of vehicle-treated animals (Fig. 5g, h)[15]. These results were confirmed by flow cytometry which showed a strong correlation between macrophage recruitment and USPIO uptake ($R^2$: 0.78, $p = 0.004$, Fig. 5i), validating the specificity of the macrophage-tracking approach[22,24].

To further characterize signatures of response and resistance by imaging, we developed an artificial neural network (ANN)-based pipeline for automated brain tumor segmentation and performed radiomic-based predictive modeling (Fig. 5j–l, Supplementary Fig. 6g)[25]. The segmentation ANN was developed using the nnUNET framework[26] and trained on our previously published work[15]. The ANN was trained to segment tumors on T2w images ($n = 188$ mice; $n = 564$ MRI exams). We then extracted 383 radiomic features from the automated ANN segmentations to predict treatment response (R vs NR) at each time point. Features were extracted from the first two MRI time points and incorporated with features derived from the change in radiomic features between the first and second time points (MRI1-MRI2). Thus, we could predict treatment response or resistance with an accuracy of 84.7% (95% confidence interval, 81.7–87.4%; sensitivity: 80.8%; and specificity: 86.9%, Fig. 5j). Predictive accuracy was significantly higher as compared to the null model (no information rate of 63.6%) ($p < 0.001$, Fig. 5j, k). The top radiomic feature for prediction of therapy failure to therapy was the difference in volumetric compactness between MRI1 and MRI2 (Fig. 5l), indicating that advanced radiomic feature and tumor texture analysis allow response prediction before therapeutic efficacy can be visualized by current morphological imaging.

## Discussion

Resistance to immunotherapies is a major challenge in oncology and seems particularly relevant in glioma immunotherapy, with only minimal efficacy seen in clinical trials, despite immunomodulatory effects on the tumor microenvironment[27–29]. T cell numbers are generally low in glioma, and except for mismatch repair-deficient gliomas, neoepitope numbers are lower in glioma compared with other solid tumors. Given the high numbers of macrophages and microglia in gliomas, composing up to 50% of the whole tumor mass[30], we evaluated specific targeting of the myeloid compartment by the TLR agonist resiquimod (R848). During glioma progression, blood-borne myeloid cells increasingly infiltrate the tumor and drive immunosuppressive programs within the TME[2]. Here we show that direct targeting of this compartment by systemic CDNP-R848 therapy is highly efficient in established syngeneic experimental glioma.

TLR agonists have been extensively investigated in several infectious and neoplastic conditions[31]. The TLR7 agonist imiquimod (R837) and the related resiquimod (R848) have been primarily tested in topical skin tumors due to systemic off-target effects. However, systemic toxicity after intravenous administration can be considerably reduced by suitable NP-packaging[9,32]. Preclinical studies have established the efficacy of R848 in the Gl261 glioma model after intratumoral injection[33]. Recently, a new targeting platform for R848 has been established by packaging R848 into a cyclodextrin nanoparticle (CDNP-R848) by supramolecular interactions. This showed good bioavailability and specific targeting of tumor myeloid cells without signs of systemic toxicity. In the tumor context, this approach potently induced pro-inflammatory macrophages and served as an effective anti-cancer treatment in solid, subcutaneous tumor models when combined with ICB[15]. This effect was dependent on CD8+ effector T cells[15]. Interestingly, we found an increased clinical efficacy of monotherapeutic CDNP-R848 in the Gl261 glioma model, with an overall response rate of 83%. CDNP-R848 was preferentially taken up by monocytic derived myeloid cells and tissue-resident myeloid cells as compared to microglia. This indicates that in line with our previous work on USPIOs[22], NP uptake occurs mainly in the circulation by TAMs that get recruited to the TME, whereas direct leakage of the NP into the TME through the disrupted BBB occurs to a lesser degree.

Mechanistically, we demonstrate a profound re-shaping of the TME with changes in the chemokine and cytokine secretion even after a single dose of CDNP-R848. IL-12 was the dominant cytokine that was markedly upregulated and has been implicated as an essential regulator of cytotoxic CD8+ T cells, CAR-T cells[34], and NK cell subsets[35] in solid tumor models including melanoma[31] and glioma[36] with recently demonstrated clinical potential for glioma treatment[37]. Gliomas of CDNP-R848 treated mice depicted an altered T cell composition with increased CD4+ T cell proliferation and decreased Lag3-mediated T cell exhaustion, likely driven by IL-12 upregulation. These data are supported by evidence from previous studies describing similar changes in T cell phenotype following IL-12 therapy in subcutaneous tumor[38] and glioma models[34]. However, we found that CDNP-R848 treatment efficacy was independent of CD8+ T cells, CD4+ T cells, and NK cells since selective depletion of these immune cells did not abrogate the treatment effect. This suggests an alternative mechanism of tumor clearing induced by CDNP-R848 treatment and is in line with previous findings in a PDAC model after TLR9-targeted therapy[39].

Here, we have shown that CDNP-R848 treatment leads to a profound shift of macrophage phenotype towards a pro-inflammatory state with increased frequencies of pro-inflammatory macrophages (F4/80+/MHCII+) and decreased levels of anti-inflammatory MDSCs (Ly6C+/PD-L1+). Additionally, we observed enhanced ROS production and pro-inflammatory polarization of BMDMs following CDNP-R848 treatment. Notably, co-culturing of BMDMs with glioma cells led to increased tumor cell death in the presence of CDNP-R848. Therefore, we propose that during CDNP-R848 treatment, increased ROS production by pro-inflammatory macrophages[40,41] in conjunction with the reduced

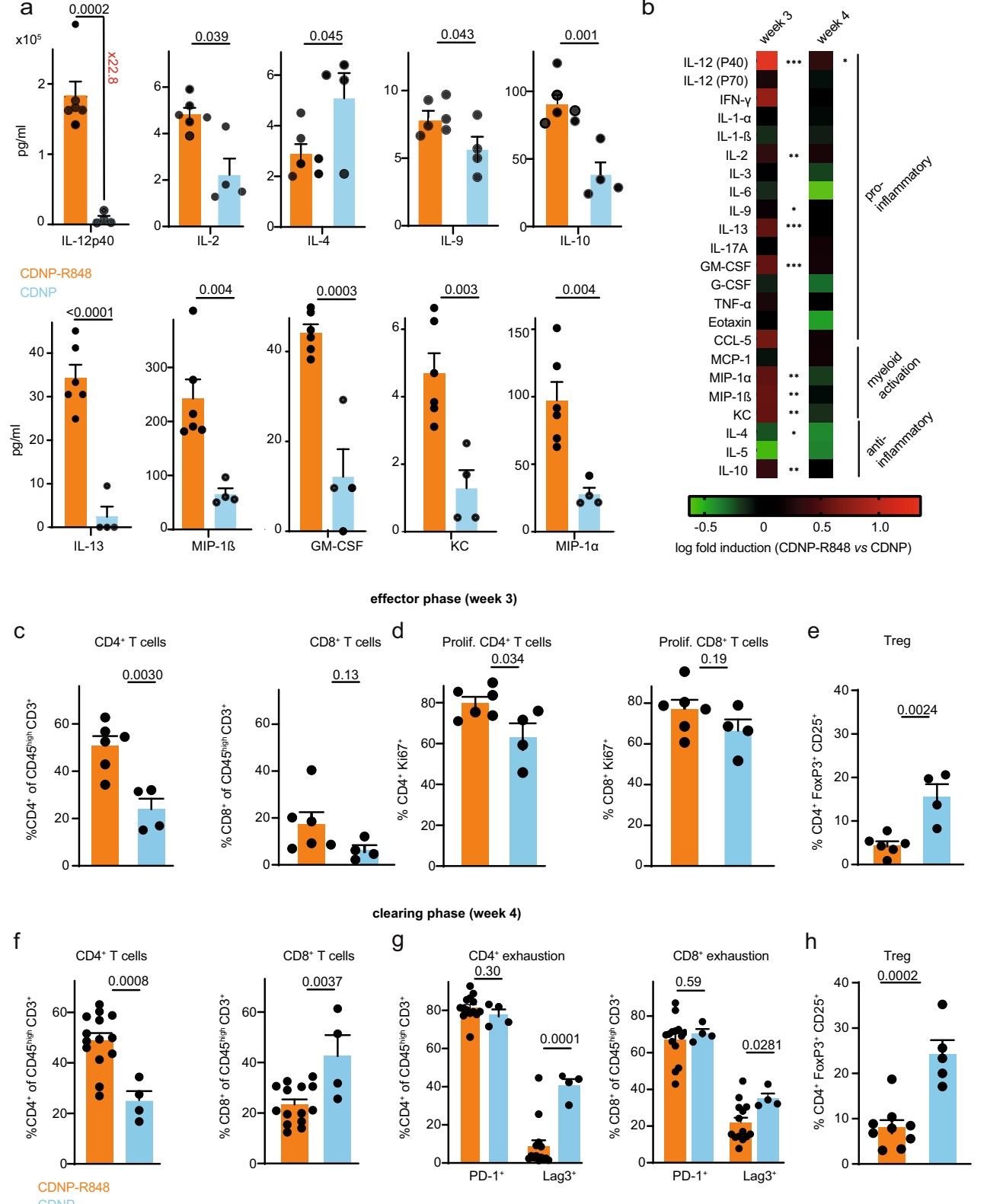

**Fig. 3 | CDNP-R848 shapes the tumor microenvironment. a** Luminex cytokine profiling at d21 (CDNP-R848: $n = 6$ mice vs CDNP: $n = 4$ mice, pooled data from two independent experiments). After enzymatic digestion of the tumor-bearing hemisphere, supernatant was taken and Luminex cytokine profiling was performed for indicated cytokines. **b** Heatmap showing Luminex cytokine results for effector (week 3) and clearing phase (week 4). **c**−**h** Flow cytometric analysis of tumor-infiltrating T cells. Gl261-bearing mice were sacrificed on d21 post-tumor inoculation (effector phase, week 3, **c**−**e** CDNP-R848: $n = 6$ mice vs CDNP: $n = 4$ mice, one independent experiment) or d27 post-tumor inoculation (clearing phase, week 4, **f**−**h**, CDNP-R848: $n = 14$ mice vs CDNP: $n = 4$ mice, two independent experiments). The data are presented as mean ± SEM. Statistical significance was determined by two-tailed Student's $t$ test. FC: fold change.

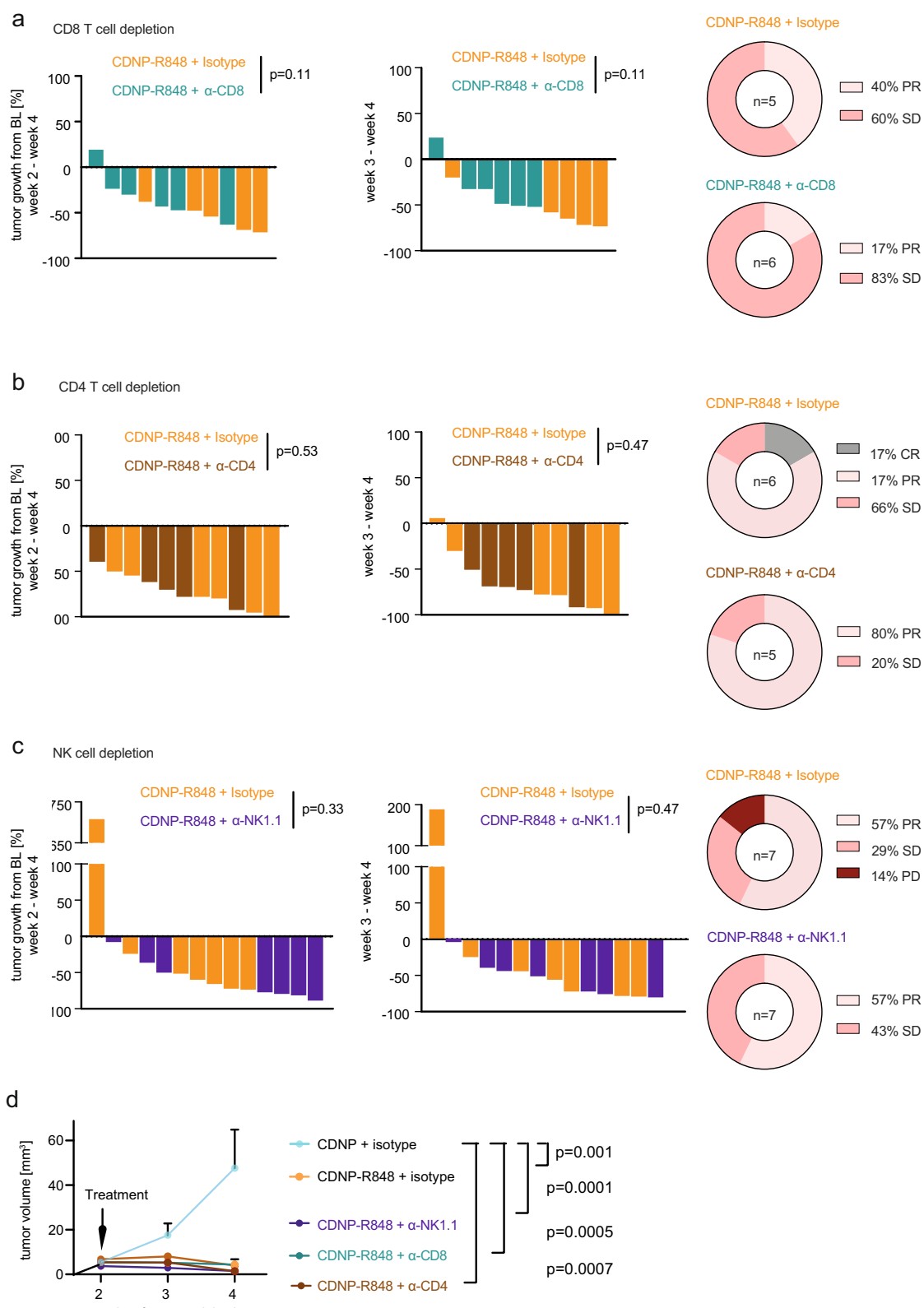

**Fig. 4 | CDNP-R848 treatment effect is independent of CD8 and CD4 T cells, and NK cells. a–c** Waterfall plot of tumor volumes (left) and response assessment (right) for immune cell depletion experiments. CD8[+] T cells, CD4[+] T cells or NK1.1[+] were depleted prior to and during CDNP-R848 therapy using monoclonal depletion antibodies. Tumor volume (% between d13 and d26) and response assessment of CD8[+] T cell-depleted or isotype treated mice (CDNP-R848 + CD8 isotype n = 5 vs CDNP-R848 + α-CD8 depleted, n = 6 animals, **a**), CD4[+] T cell-depleted or isotype treated mice (CDNP-R848 + CD4 isotype n = 6 vs CDNP-R848 + α-CD4 depleted, n = 5

mice, **b**) and NK1.1+ depleted or isotype treated mice (CDNP-R848 + NK1.1 isotype, n = 7 vs CDNP-R848 + α-NK1.1 depleted, n = 7 animals, **c**). **d** Tumor growth of the different treatment and depletion groups (α-NK1.1 depleted, n = 7 animals; α-CD8 depleted, n = 6 animals, α-CD4 depleted, n = 5 mice; CDNP-R848 + isotype n = 18 mice; CDNP + isotype, n = 4 mice). The data are from one independent experiment and presented as mean ± SEM. Statistical significance was determined by two-tailed Student's t test.

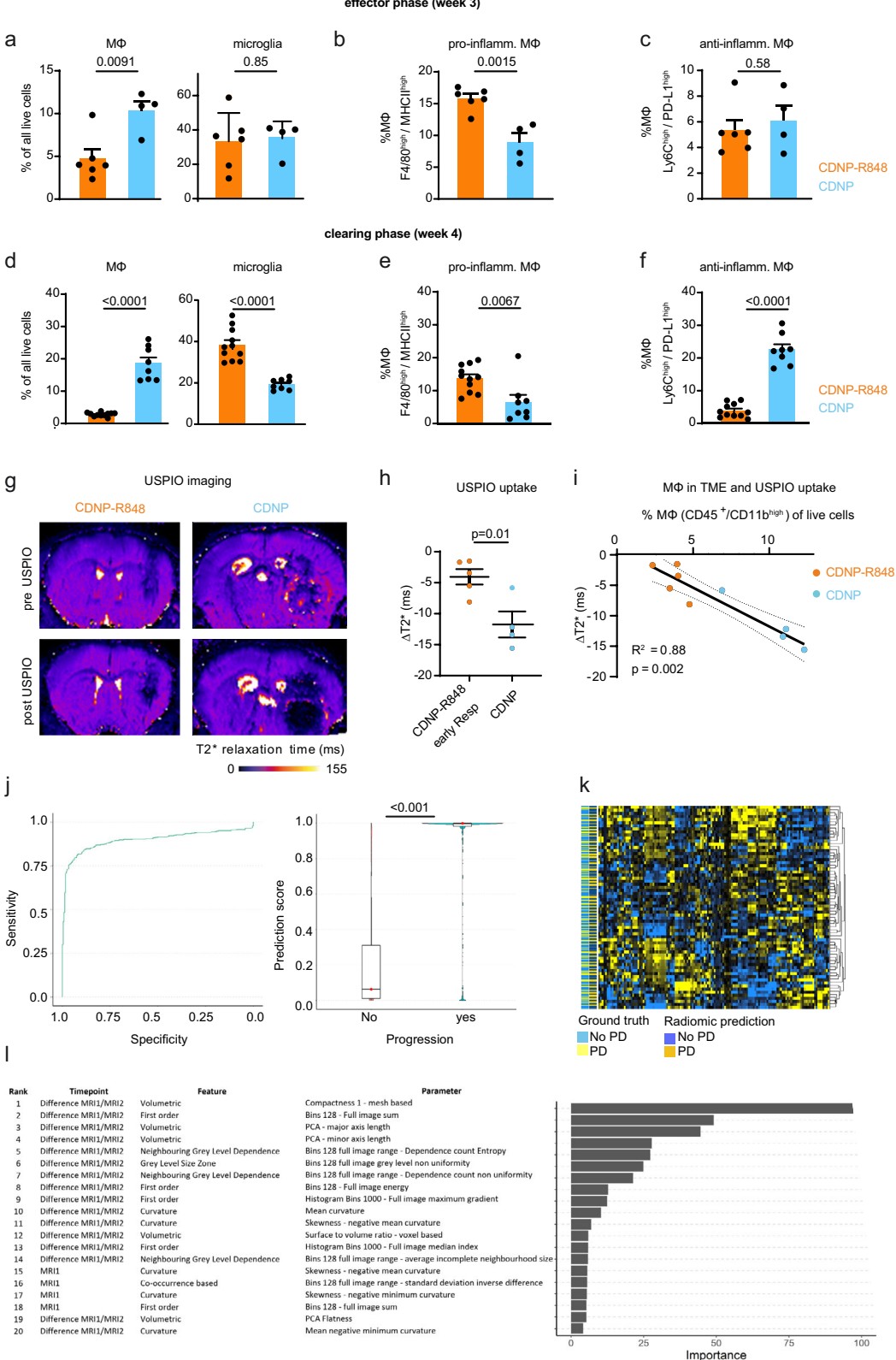

recruitment of tumor-promoting MDSCs underlies the strong mono-therapeutic treatment effect in our glioma model (Supplementary Fig. 6h).

In general, NP formulations have emerged as important carriers for drug formulations and diagnostic compounds for companion diagnostics[42]. We demonstrate that non-invasive macrophage imaging by iron oxide nanoparticles can be used to monitor CDNP-R848

therapy. This seems particularly relevant because immunotherapy is characterized by response, resistance, and pseudoprogression, which we model in our treatment paradigm. We show that unlike conventional MRI sequences (diffusion imaging, Gd-contrast enhanced imaging), phagocyte-specific MRI can visualize an increased influx of myeloid cells in non-responder animals, thus predicting resistance to therapy. Establishing imaging markers of immunotherapeutic effects

**Fig. 5 | Macrophage- and radiomics-based multiparametric MRI signatures of CDNP-R848 therapy response. a** Quantification of macrophages (CD45[+], CD11b[high]) and microglia (CD45[+], CD11b[int]) in the effector phase (week 3, d21); **b** Quantification of MHC-II and **c:** PD-L1 on TAMs and MDSCs in the effector phase (**a**–**c**: $n = 6$ mice for CDNP-R848; $n = 4$ mice for CDNP, one independent experiment) **d** Quantification of macrophages/microglia; and **e:** MHC-II and **f:** PD-L1 on macrophages in the tumor-clearing phase (week 4, day 28; **d**–**f**: $n = 11$ mice for CDNP-R848 and $n = 8$ mice for CDNP, pooled data from two independent experiments). **g** Representative magnetic gradient echo (MGE) maps before and 24 h after USPIO administration (ferumoxytol, 30 mg/kg, representative images from 9 independent mice). USPIO imaging was performed on d20 after completion of treatment. **h** Quantification of USPIO accumulation in CDNP-R848 early responder mice vs CDNP vehicle-treated animals (CDNP-R848, $n = 5$ mice; CDNP, $n = 4$ mice, one independent experiment). **i** Correlation analysis of $\Delta$ T2* relaxation time and

macrophage influx as quantified by flow cytometry in CDNP-R848 responder mice vs CDNP vehicle-treated animals on d21. (CDNP-R848; $n = 5$ vs CDNP; $n = 4$ mice from one independent experiment). **j** Radiomic signature prediction score across the datasets for predicting treatment response based on radiomic features of MRI1 (baseline, week 2) and MRI2 (week 3 after completion of the therapy cycle). ROC curve for the performance of the developed gradient boosting model. ($n = 66$ animals with 3 MRI datasets each from 5 independent experiments). Boxplot with blocks showing the interquartile range (IQR) of data points and the horizontal central line (red dot) corresponds to the median. The superimposed violin plot visualizes the distribution of the data and its probability density. **k** Heatmap of radiomic features. **l** Top predictive radiomic features for the developed model. The data are presented as mean ± SEM. Statistical significance was determined for two groups by two-tailed Student's *t* tests or by one-way ANOVA for multiple group comparisons. Correlation analysis was performed by Spearman correlation.

beyond tumor size is of utmost importance and highlighted in our work.

Furthermore, we use predictive modeling based on radiomic features to predict therapy response with high accuracy within our cohort. Our findings are in line with our previous study on checkpoint blockade[15]. Notably, although only 20% of the top features (classified by importance within the model) were shared between the radiomic signatures, many of them belonged to the same feature sub-categories, hinting at the possibility of developing a shared radiomic signature that is independent of the underlying treatment mechanism and which could support clinical response assessment as recently shown for antiangiogenic agents[43].

In summary, our study reveals a treatment paradigm in a syngeneic glioma model using the CDNP-R848 platform, which activates and re-educates the immunosuppressive myeloid compartment and acts independently of the adaptive immune system as well as NK cells. This underlines that the suppressive myeloid compartment in glioma can be specifically targeted, thus paving the way for new, druggable targets for glioma immunotherapy that can be assessed by phagocyte-directed imaging.

# Methods
## Mice
Specific and Opportunistic Pathogen Free (SOPF) female C57Bl/6J mice were bought from Janvier Laboratories at the age of 6-10 weeks and used for experiments one week later. All animal protocols were performed in compliance with the laboratory animal research guidelines and were approved by the governmental authorities (animal protocols: G27-17 and G35-22, regional administrative authority, Regierungspräsidium Karlsruhe, Germany). Experimental and control animals were co-housed at the specific pathogen-free/SPF animal facility of the DKFZ Heidelberg

## Cell lines
Gl261 cells were purchased from the National Cancer Institute Tumor and cultured in Dulbecco's modified Eagle's medium (DMEM) supplemented with 10 % fetal bovine serum (FBS) and 100 U/ml penicillin and 100 µg/ml streptomycin (all Sigma-Aldrich) at 37 °C, 5% $CO_2$. Gl261 cells were routinely tested for contamination by multiplex cell contamination test (Multiplexion GmbH). Gl261 cells were not passaged more than ten times.

## Tumor cell inoculation
$1 \times 10^5$ Gl261 tumor cells were resuspended in 2 µl sterile PBS (Sigma-Aldrich) and stereotactically implanted into the right basal ganglia of 7-11 weeks old female C57Bl/6J mice (coordinates: 2 mm right lateral of the bregma and 1 mm anterior to the coronal suture with an injection depth of 3 mm below the dural surface) using a 10 µl Hamilton microsyringe driven by a fine step stereotactic device (Stoelting) as previously described[15].

## CDNP synthesis and treatment
β-Cyclodextrin nanoparticles (CDNPs) were prepared and formulated with drug as previously described by Rodell et al.[9]. Briefly, a scintillation vial was charged with succinyl-β-cyclodextrin, *N*-(3-dimethylaminopropyl)-*N'*-ethylcarbodiimide hydrochloride (EDC, 10.0 eq. relative to succinate groups), and *N*-hydroxysuccinimide (NHS, 5.0 eq.) and dissolved in MES buffer (50 mM, pH 6) by stirring. After 30 min, *L*-lysine (0.5 eq.) was added via dropwise addition to yield a final concentration of at 3.3%$_{w/v}$ succinyl-β-cyclodextrin. After overnight reaction at ambient conditions, the product was purified by precipitation from a ten-fold excess of iced ethanol, size-exclusion chromatography (PD-10, Fisher), and subsequent concentration by centrifugal filtration (10 kDa MWCO, Amicon). Particle size ($D_{z-ave}$: 20.7 ± 1.3 nm; PDI = 0.66 ± 0.06) was determined by dynamic light scattering (DLS; Zetasizer, Malvern), and products were endotoxin tested prior to use (RAW-Blue™ assay, according to manufacturer protocols). For assessing biodistribution, CDNPs (20 mg/mL) were fluorescently labeled by VivoTag 680 XL (Perkin Elmer, 50 µM) in carbonate buffer (100 mM, pH 8.5), with product recovery by centrifugal filtration (10 kDa MWCO, Amicon) and repeated washed to remove unbound fluorophore. Lyophilized nanoparticles were dissolved at 50 mg/mL in MilliQ water and stored at −20 °C until use. A dose of 0.2 mg (10 mg/kg) R848 and 2.25 mg CDNP (100 µL of R848 (Sigma), CDNP, or equivalent dosing of CDNP-R848) per mouse were administered by intravenous tail vein injection on days 14, 17 and 20 after tumor inoculation. 100 µl of phosphate buffered saline (PBS) was used as an additional control group.

## In vivo depletion experiments
Depletion of CD8[+] and CD4[+] T cells, and NK cells was performed by i.p. injection of respective in vivo antibodies in 200 µl PBS per mouse (all from BioXCell): anti-CD4 (GK1.5, 1000 µg/mouse two injections 12 and 19 days after tumor inoculation); anti-CD8 (2.4, 500 µg /mouse, four injections 12, 18, 21, and 25 after tumor inoculation); anti-NK 1.1 (200 µg /mouse, four injections 11, 15, 19, and 23 days after tumor inoculation). Isotype controls (clone LTF2 for anti-CD4 and anti-CD8 depletion; clone C1.18.4 for anti-NK 1.1 depletion) were administered at the same respective dose. Immune cell depletion was performed prior to CDNP-R848 treatment to ensure effective depletion before initiation of treatment. Successful depletion was confirmed by peripheral blood analysis prior to and during treatment. At the experiment endpoint, depletion was verified at the brain tumor site using flow cytometry of the respective immune cell population in the TME (for CD4 and NK) or immunohistochemistry (for CD8) (Supplementary Table 1).

## MRI
Multiparametric MRI was performed on a 9.4 T horizontal bore small animal NMR scanner (BioSpec 94/ 20 USR, Bruker BioSpin GmbH, Ettlingen, Germany) with a four-channel phased-array surface receiver coil. MR imaging included a standard RARE T2-w and T1-w post-Gd-contrast sequence to monitor tumor volume (T2-w parameters: 2D

sequence, 0.078 mm in-plane resolution, TE: 33 ms, TR: 2500 ms, flip angle: 90°, acquisition matrix: 200 × 150, number of averages: 2, slice thickness: 0.7 mm, duration: 2 min 53 s; T1-w parameters: 2D sequence, 0.1 mm in-plane resolution, TE: 6 ms, 1000 TR: ms, flip angle: 90°, acquisition matrix: 256 × 256, number of averages: 2, slice thickness: 0.5 mm, duration: 5 min). Further functional MRI included diffusion tensor imaging (parameters: 2D EPI sequence, 30 diffusion gradient directions, 0.125 mm in-plane resolution, TE: 20 ms, TR: 3400 ms, flip angle: 90°, acquisition matrix: 96 × 96, number of averages: 1, slice thickness: 0.7 mm, duration: 7 min 56 s) and multi-gradient echo imaging (MGE parameters: 3D sequence, 0.1 mm in-plane resolution, TE: 2.57 ms, TR: 73.43 ms, flip angle: 20°, acquisition matrix: 200 × 200, number of averages: 2, slice thickness: 0.1 mm, duration: 22 min 7 s). As contrast agent 0.2 mmol/kg Dotarem (Guerbet) was administered i.v. to assess BBB integrity with T1-w. MGE imaging was used for macrophage tracking using the ultrasmall superparamagnetic iron oxide (USPIO) nanoparticle ferumoxytol (Feraheme; AMAG Pharmaceuticals Inc.). Imaging was performed before and 24 h after ferumoxytol (dose of 30 mg/kg).

For all MRI procedures, animals were anesthetized with 3% isoflurane. Anesthesia was maintained with 1–1.5% isoflurane. Animals were kept on a heating pad to maintain constant body temperature, and respiration was monitored externally during imaging with a breathing surface pad controlled by a LabVIEW program developed in house (National Instruments Corporation).

### Analysis of MRI data

MR images were exported as DICOM files and were visualized in OsiriX Imaging software (version 4.12; Pixmeo) and FIJI (FIJI/ImageJ, Version 2.0). For the quantification of MRI data, tumor volumes were segmented semi-automatically using AMIRA (FEI, version 5.4).

T2*-relaxation times were calculated from MGE raw data and exported as T2*-relaxation maps in DICOM-format using a customized script (MATLAB R2020a, 64-bit version for Windows, MathWorks). The diffusion tensor was calculated from diffusion data using Bruker's ParaVision Software V.6.0 and maps of fractional anisotropy (FA) were exported in DICOM-format. The FA-maps were displayed in FIJI and a region of interest was placed centrally in the tumor. Mean FA was extracted and used for statistical analyses. T2*-maps were quantified using open-source Slicer Software (3D Slicer V. 4.11.0, www.slicer.org)[44]. Tumor volume was segmented semi-automatically using Slicer's Segment Editor module. Mean T2*-relaxation time of the entire tumor volume was extracted for further analyses.

### Radiomic analysis

Radiomic analysis of MRI data was performed with an established workflow as described previously[15]. Briefly, lesion volumes (MRI1/week 2: baseline lesion volumes, MRI2/week 3: treatment volumes during effector phase, and MRI3/week 4: lesion volumes during clearing phase) were segmented on T2-weighted MR images using a fully automated artificial neural network developed from the nnU-NET framework[26] and trained on our previously MRI data ($n = 188$ mice; n = 564 MRI exams)[15]. Mice were randomly assigned to a training set (161 mice with a total of 483 MRI scans) and test set (27 mice with a total of 81 MRI scans). Tumors were manually segmented from a human rater at each time point using ITK-SNAP (www.itksnap.org). The performance of the ANN was evaluated in the training set (with 5-fold cross-validation (CV)) and the test set using the DICE similarity coefficient (for tumor segmentation agreement) score and the concordance correlation coefficient (CCC, for tumor volume agreement). We obtained a median DICE coefficient score of 0.866 (IQR 0.816-0.899) in the training set and 0.881 (IQR 0.816-0.908) in the test set. The CCC of the segmentations resulted in excellent agreement in both sets, with mean scores of 0.982 (95% CI, 0.978-0.985) for the training set and 0.988 (95%CI,

0.981-0.992) for the test set. Images were skull stripped and normalized with Z-score normalization using FSL (FMRIB, Oxford). We then extracted a set of 383 radiomic features from the automated ANN segmentations to predict treatment response (R vs NR) at each time point using MITK (DKFZ, www.mitk.org). Prediction models were developed using R (Version 4.0.3, Foundation for Statistical Computing, Vienna) and the caret package[45] with the use of Gradient Boosting Machines (GBMs). The performance of the gradient boosting classifier was assessed based on a ten-times repeated ten-fold cross-validation resampling procedure. Predictive modeling was performed using the GBMs machine-learning algorithm that iteratively constructs an ensemble of weak decision tree learners through boosting to form a single strong predictive model (the tuning parameters [boosting iterations, max tree depth, shrinkage, and min. terminal node size] were automatically optimized via resampling procedures). The held-out predictions in each of the resampling iterations were used to calculate the accuracy, area under the receiver operating characteristic (ROC), sensitivity, specificity, no information rate (largest class percentage for each molecular parameter, i.e., the prediction or accuracy by chance), and a hypothesis test (using the binom.test function) to evaluate whether the accuracy rate is greater than the no information rate. $P$-values < 0.05 were considered significant. See supplementary notes for additional information on the radiomic features.

### Tumor response criteria and survival analysis

Tumor response criteria were based on weekly MR imaging. For tumor monitoring, MRI imaging was performed 2 weeks (baseline, day 13), 3 weeks (day 19) and 4 weeks (day 26) after tumor cell inoculation. Classification of tumor response was assessed as previously published[15]. CR was defined as a complete regression of tumor volume MRI1–MRI3 (%VMRI3–MRI1) of −100%, PR as % VMRI3–MRI1 ≤ − 65.0% and/or %VMRI3–MRI2 ≤ − 65.0%, SD as % VMRI3–MRI1 > − 65% and < + 40%, and PD as %VMRI3–MRI1 ≥ + 40%. Mice with unconfirmed progression between MRI2 and MRI3 were defined as SD, if tumors regressed at least 30% between MRI2 and MRI3 (%VMRI3–MRI2 ≤ − 30.0%). Mice with CR, PR, or SD were defined as responder mice. PsPD was defined as an increase in tumor volume of >40% compared to baseline in week 3, followed by tumor regression[46]. Early response was defined as a tumor decrease compared to baseline in week 3. For survival analysis mice were randomized to either 3 doses of CDNP-R848 or vehicle treatment using the same treatment scheme as described above (100 μl of CDNP-R848 or equivalent dose of CDNP vehicle control on days 14, 17 and 20 after tumor inoculation). Animals were imaged weekly in the following weeks and scored daily for possible neurological symptoms. Animals were euthanized by receiving a lethal dose of ketamine/xylazine when termination criteria were fulfilled or when the tumor size reached a volume of 100 mm³ (maximum tumor size).

### Isolation of splenocytes, blood leukocytes and tumor-infiltrating leukocytes

After euthanizing, spleens were excised and meshed twice through a 70 μm cell strainer to obtain a single-cell suspension. Red blood cells were lysed with ACK buffer (150 mM $NH_4Cl$, 10 mM $KHCO_3$, and 100 μM $Na_2EDTA$). For isolation of tumor-infiltrating leukocytes, mice were cardially perfused after receiving a lethal dose of ketamine/xylazine. The cerebellum was removed, and the tumor-bearing hemisphere was separated from the non-tumor-bearing hemisphere and processed separately. After enzymatical digestion (HBSS, Sigma (Aldrich) with 50 μg/ml Liberase D (Roche) for 30 min at 37°, the tissue was meshed twice through 100 μm and 70 μm cell strainers to obtain a single-cell suspension. Myelin removal was done with a 30% Percoll gradient (GE Healthcare, Princeton, NJ, USA) according to the manufacturer's instruction.

## Flow cytometry

Brain tumor and spleen cell suspensions were blocked with anti-CD16/CD32 (eBioscience; 93; 14-0161). BMDMs were incubated with Fc-γ receptor blocking solution and the viability staining solution DAPI (SBA-0100-20, Biozol). Extracellular targets were stained for 30 min at 4 °C using the antibodies listed in (Supplementary Table 2). For intracellular cytokine staining, cells were incubated with 5 μg/ml Brefeldin A (Sigma-Aldrich) for 5 h at 37 °C, 5% $CO_2$ to allow for intracellular enrichment of cytokines. Cells were subsequently fixed, permeabilized, and stained using the FOXP3/transcription factor staining buffer set (eBioscience; 00-5523). Staining of intracellular targets was performed for 45 min at 4 °C. Stained cells were analyzed on FACS Aria II, LSRFortessa (both BD Biosciences; Germany), AURORA spectral flow cytometer (Cytek Biosciences), or on Attune NxT (Thermo Fisher; Germany) and data were acquired with the BD FACSDIVA software V9.0 or Attune NxT Software V.2.5. MFI is defined as the geometric median fluorescence intensity and was calculated by subtracting the MFI of the cells stained with the isotype-matched antibody from the MFI of those stained with the specific antibody and is shown as fold-change compared to the non-treated (NT) control. FlowJo V9 or V10 were used for data analysis.

## Immunohistochemistry

For histological correlation analysis, mice were killed in deep anesthesia by intracardial perfusion with PBS. Brains and spleens were dissected, cut and freshly mounted as thick sections with DAPI or snap frozen in Tissue-Tek® O.C.T.TM (Sakura). Staining for CD8 + T cells was performed in 1:50 dilution of PE-conjugated anti-CD8 antibody (Thermo Fischer; 12-0081-82 or mCD8, PE, 56-6.7, Invitrogen, 12-0081-82, LOT: 2207581). In brief, cryo-sections were fixed with ice-cold acetone (−20 °C) for 10 min. After washing, slides were blocked with SuperBlock™ T20 (TBS) Blocking Buffer (Thermo Fisher) for 30 min at room temperature (RT). CD8 staining was performed in TBS for 1 h at RT. Tile scans (10x of the entire tumor-bearing hemisphere or spleen) and higher-magnification images (40x) were acquired by confocal microscopy (Zeiss LSM700) using the ZEISS ZEN software version 2.3.

## Biodistribution of fluorescently labeled CDNP-R848 (CDNP-R848-VT680)

Gl261-bearing mice were treated with a single intravenous dose of CDNP-R848-VT680. After 24 h, distribution was assessed using immunohistochemistry of the tumor-bearing hemisphere or flow cytometry of myeloid cells as described above.

## Analysis of standard blood parameters

Gl261-bearing mice were treated with CDNP-R848, CDNP, R848, or PBS following the above-mentioned. During the experiment, blood from individual mice was collected in EDTA-coated tubes by puncture of the facial vein. At the end of the experiment, blood was taken by terminal heart puncture. Blood was stored on ice until data acquisition was conducted. Analysis of blood samples was performed with Scil Vet abc Plus+ analyzer (Covetrus) according to the company's protocol.

## Luminex mouse cytokine multiplex assay

The mouse cytokine multiplex-23 bead array assay kit for Luminex (Bio-Rad) was used to measure the following cytokines in the tumor-bearing hemisphere: IL-1α, IL-1β, IL-2, IL-3, IL-4, IL-5, IL-6, IL-7, IL-8, IL-10, IL-12p40, IL-12p70, IL-13, IL-15, IP-10, MCP-1, MIP-1α, CCL-5 (RANTES), TNF-α, IFN-α2, IFN-γ, GM-CSF and Eotaxin (CCL-11) according to the manufacturer's protocol. Supernatants of homogenized Gl261-bearing C57Bl/6J tumor hemispheres after single or triple dose treatment of CDNP-R848 or CDNP vehicle control were used undiluted. Supernatants of cultured Gl261 cells incubated with CDNP-R848 for 48 h were analyzed using a dilution of 1:10 or 1:5. The assay was performed in 96-well Bio-Plex Pro Flat Bottom Plates. For analysis, Bio-Plex 200 System (Bio-Rad) was used and cytokine standards supplied by the manufacturer were used as controls using Bio-Plex Manager software (version 6.0).

## In vitro BMDM and tumor cell co-culture

Generation of BMDM cultures was performed according to a previously established protocol[47]. Briefly, BM cells were flushed from the tibia and femurs of C57BL/6 N wild-type mice (8-10 weeks of age) using ice-cold Hanks' Balanced Salt Solution (HBSS) and filtered through a 70 μm cell strainer and plated at a density of $3.5 \times 10^5$ cells/ml. Cells were differentiated for one week using RPMI medium supplemented with 10 ng/ml M-CSF (M9170, Sigma-Aldrich), 10% fetal bovine serum (FBS) and 1% penicillin/streptomycin (Gibco). For each independent experiment, BMDMs were prepared from three different mice.

## ROS production of BMDMs

Accumulation of ROS in BMDMs was assessed by using the oxidant-sensitive fluorescent probe CellROX™ Green Reagent (Life Technologies). Upon cellular uptake, the non-fluorescent CellROX™ probe undergoes deacetylation by intracellular esterases producing a highly green fluorescent signal following oxidation by intracellular ROS. BMDMs were maintained untreated or treated for 24 h with 100 ng/mL lipopolysaccharide (LPS), 1 μM CDNP-R848 or CDNP. 2.5 mM of CELLROX™ was added to cells and incubated for 30 min at 37 °C under 5% $CO_2$ atmosphere. Cells were washed twice with HBSS and fluorescence intensity was measured by flow cytometry. Fluorescence intensity is represented as fold change compared to the non-treated condition (NT).

## Real-time cell analysis

To assess the possible cytotoxic effect of CDNP-R848 on Gl261 cells, real-time cell analysis (RTCA, xCELLigence) was performed. Before seeding, background measurements were taken from the wells by adding 50 μl of DMEM medium supplemented with 10% fetal bovine serum (FBS) and 100 U/ml penicillin and 100 μg/ml streptomycin to E-16plates. Afterwards, Gl261 cells (10,000 cells/well) were added and incubated at 37 °C and 5% $CO_2$ for 24 h. Tumor cells were treated with increasing amounts of CDNP-R848 or CDNP vehicle control (1 nM to 10 μM) and cell proliferation was assessed for 72 h. During the experiment cell attachment was measured every 15 min by the xCELLigence device. Cell index (CI) for real-time dynamic cytotoxicity assessment was calculated within the RTCA Software Package version 1.2.1. All experiments were run as technical triplicates (For primer sequences, see Supplementary Table 3).

## qRT-PCR

Total RNA was extracted from cells using the RNeasy Mini Kit (74134, Qiagen). 1 μg of total RNA was reverse transcribed by using RevertAid H Minus Reverse Transcriptase (FERMEP0452, Thermo Scientific), random primers (48190-011, Invitrogen) and dNTPs (R0193, Thermo-Scientific). qRT-PCR was performed on a Step One Plus Real-Time PCR System (Applied Biosystems, California, USA). Primers and probes were designed using the ProbeFinder software (www.roche-applied-science.com). Gene expression was normalized to the housekeeping gene *Rpl19* and differences in Relative Quantity (RQ) are shown as fold-change compared to the control condition (untreated cells, NT).

## Cytotoxicity

GL261 cell viability was quantified using the CytoTox 96 kit (Promega). Briefly, GL261 cells and BMDMs or BMDMs alone were plated in a black side/black bottom 96 well plate at a ratio of 1:2 to a maximum of 15,000 total cells in 100 μL per well. To measure LDH release into the supernatant, the plate was centrifuged at 500 g for 10 min to sediment cells. 50 μL of supernatant was used and added to 50 μL of substrate and subsequently incubated for 30 min at room temperature in the dark. After 30 min, 20 μL stop solution was added to each well and

absorbance was measured at 490 nm by a spectrofluorometer (SpectraMax, Molecular Devices). Cytotoxicity was calculated as per manufacturers instructions by first subtracting the media blank value from experimental values and then applying the Cell-mediated Cytotoxicity formula. All values are represented as fold change with respect to the non-treated condition.

### Single-cell analysis of Gl261-infiltrating immune cells

Single-cell analysis of TLR7/8 and downstream targets was performed using the data and platform which was provided by Pombo Antunes et al.[10].

### Statistics

Data are represented as individual values or as mean ± SEM. Group sizes (n) and applied statistical tests are indicated in figure legends. Significance was assessed by either unpaired t-test analysis, paired t-test analysis, or one-way analysis of variance (ANOVA) analysis with Tukey post hoc testing as indicated in figure legends. Spearman's correlation was applied for all correlation analyses, and the Kaplan–Meier method was used to examine survival. Statistics were calculated using GraphPad Prism 7.0.

### Reporting summary

Further information on research design is available in the Nature Portfolio Reporting Summary linked to this article.

## Data availability

The raw numbers for charts and graphs are available for Figs. 1a–d, 2d, f, 3b, 4a–d, 5h and Supplementary Figs. 3a, f, 4b, 5b–d, 6d in the Source Data file. Additional information for radiomic feature predictions for Fig. 5j–l is included in the Supplementary notes. Additional data are available from the authors upon reasonable requests, as are unique reagents used in this article. Source data are provided with this paper.

## Code availability

Custom-written code is publicly available: The CNN for radiomic response prediction is available at github.com (https://github.com/NeuroAI-HD/HD-GLIOMOUSE). The radiomic brain extraction was performed with MITK (www.mitk.org; https://phabricator.mitk.org/w/mitk/changelog/release-v2022.10). For further information on software package versions please refer to the Nature Research Reporting Summary linked to this article.

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

## Acknowledgements

The authors acknowledge support from the Core Facility for Flow Cytometry (Dr. Steffen Schmidt), the Light Microscopy Core Facility (Dr. Damir Krunic) and small animal imaging core facility (Dr. Manfred Jugold) at the German Cancer Research Center. M.O.B. was supported by the Emmy Noether program of the German Research Foundation (DFG, BR 6153/1-1) and the Else Kröner-Fresenius Stiftung (2017-A25; 2019_EKMS.23). V.T and K.P were supported by a fellowship from the Mildred-Scheel doctoral program of the German Cancer Aid. T.B., W.W., P.V., D.N., M.S., M.B., M.P., and M.O.B. were supported by the Deutsche Forschungsgemeinschaft (German Research Foundation, DFG)—Project ID 404521405, SFB1389—UNITE Glioblastoma, WP A03, B01, B03, C02, C03 and D02, Project ID 394046768. M.P. was further supported by grants from Dr. Rolf M. Schwiete Foundation and the Sonderförderlinie 'Neuroinflammation' of the Ministry of Science of Baden Württemberg; the German Ministry of Education and Science (National Center for Tumor Diseases Heidelberg NCT 3.0 program 'Precision immunotherapy of brain tumors' and the DKTK program); the Deutsche Forschungsgemeinschaft (German Research Foundation, DFG)—Project ID 404521405 (SFB1389—UNITE Glioblastoma, WP B01), Project ID 394046768 (SFB1366—Vascular Control of Organ Function, WP C01), Project ID 259332240 (RTG2099, Hallmark of Skin Cancer, P14) and received project funding for "regulation of tumor immunity through the integrated stress response (ISR) in myeloid cells" by the German Cancer Aid (70113515). This work was further supported by the DFG; Project ID 259332240 (RTG2099, Hallmark of Skin Cancer, P14) to T.B.

## Author contributions

V.T., K.P., W.W., S.H., T.B., M.O.B., and M.P. designed the experiments. V.T., K.P., J.H., K.K.J., K.J., K.S., S.A., M.F., D.A.A., and M.O.B. performed in vivo and ex vivo experiments and analyzed data. K.P., M.F., K.S., K.K.J., Y.S., V.S., and M.O.B. performed MRI experiments and data analysis. P.V., M.B., and G.B. established the radiomic signature and performed radiomic response prediction. D.N. and M.S. supported sequencing data analysis. S.S.S., A.A., R.W., and C.B.R. synthesized, and characterized CDNP-R848 NP. N.K.H., S.A., and M.U.M. performed in vitro BMDMs experiments and toxicity analysis. V.T., K.P., M.O.B., and M.P. wrote the manuscript with input from all co-authors.

## Funding

## Competing interests

The authors declare no competing interests.

## Additional information

[1]Clinical Cooperation Unit Neuroimmunology and Brain Tumor Immunology, German Cancer Consortium (DKTK) within the German Cancer Research Center (DKFZ), 69120 Heidelberg, Germany. [2]Department of Neurology, Medical Faculty Mannheim, Mannheim Center for Translational Neurosciences, Heidelberg University, Theodor-Kutzer-Ufer 1-3, Mannheim, Germany. [3]Neuroradiology Department, University Hospital Heidelberg, 69120 Heidelberg, Germany. [4]Faculty of Biosciences, Heidelberg University, Heidelberg, Germany. [5]Department of Pediatric Oncology, Hematology and Immunology, University Hospital, Heidelberg, Germany. [6]Molecular Medicine Partnership Unit (MMPU), Heidelberg University, European Molecular Biology Laboratory (EMBL), Heidelberg, Germany. [7]Junior Research Group Bioinformatics and Omics Data Analytics, DKFZ, Heidelberg, Germany. [8]School of Biomedical Engineering, Science and Health Systems, Drexel University, Philadelphia, PA 19104, USA. [9]Biomedical Informatics, Data Mining and Data Analytics, Faculty of Applied Computer Science and Medical Faculty, University of Augsburg, Augsburg, Germany. [10]Center for Systems Biology, Massachusetts General Hospital, Boston, MA 02114, USA. [11]Department of Radiology, Massachusetts General Hospital and Harvard Medical School, Boston, MA 02114, USA. [12]Clinical Cooperation Unit Neurooncology, DKTK within DKFZ, Heidelberg, Germany. [13]Department of Neurology, National Center for Tumor Diseases (NCT), Heidelberg University Hospital, Heidelberg, Germany. [14]These authors jointly supervised this work: Michael O. Breckwoldt and Michael Platten. ✉e-mail: michael.breckwoldt@med.uni-heidelberg.de; m.platten@dkfz-heidelberg.de

