## [Peer Review File · Nature Communications]

This manuscript has been previously reviewed at another journal that is not operating a transparent peer review scheme. This document only contains reviewer comments and rebuttal letters for versions considered at *Nature Communications*.

REVIEWER COMMENTS

Reviewer #1 (expertise in nanomedicine, cancer therapy and drug delivery):

The results of this manuscript are of potential interest, but fundamental rationale for the observed effect of CDNPs, rather than physiological and in vivo response, are required:

The tridimensional structure of CDNPs is not straightforward, and the incorporation of R848 is not clear. Also, the role of L-lysine must be explained. Being of particular relevance the findings that nanoparticles improved the readouts, polymer-base or micellar-base structure of the nanocarrier must be elucidated.

Proinflammatory effects described in Figure 1 lack proper controls of the free R848, not encapsulated. One is missing the direct effect of the active molecules.

The targeting capacity of CDNP-R848 to glioma-associated macrophages and microglia is claimed, but the mechanistic justification is unclear. Likewise, crossing BBB by 20 nm structures is not trivial, and the transport mechanism is absent from the discussion.

Reviewer #2 (expertise in neuro-oncology, cancer therapy):

Overall, I think this is certainly an interesting paper that was a pleasure to read

Here are some points that I would like to raise:

- In response to many of the comments where the authors were asked to compare CDNP-R848 with R848, the group did two experiments, they showed an unremarkable spider plot and one bar plot showing tumor growth differences at week 7. Based on the data, I am not sure if this is an effect of CDNP-R848 or just an increased accumulation of R848 in the tumor. A telling experiment might be to inject R848 directly into the brain to see if the effect is purely a result of targeting or if there is an added advantage to using this complex.

- I completely agree with Reviewer 1, Comment 5 about the use of Abs to deplete the immune cells in the periphery. The authors mention having data in supplementary material that I can not see but can not comment on but I find it really surprising that this is what they found to be the case. If this were truly as clean as the authors are suggesting.

It would have instead been more interesting to see if these results would hold up in a RAG KO mouse background which is what I have seen more often in these types of brain cancer models. Since the authors claim that their novelty is that their mechanism is T cell and NK cell-independent, it would be important to do the experiments in as clean a system as possible.

- Based on the data, is there any chance that the results are due to R848 accumulation in a small compartment leading to immune-independent effects and purely a result of concentration-dependent toxicity?

- One of the largest limitations aside from those comments already addressed is that this is a study done in GL261 tumors. I find it suspicious that the group would not publish at the very least with CT2As which makes me wonder if their effects are model specific. The Weissleder used MC38s in their experiments. The fact that the authors are using GL261s makes me wonder if there is a PD-1 antagonism play that would not hold up in true GBM tumors.

Pros:

- I think the methods here to evaluate tumor responses are really remarkable and commendable. The fact that they were able to incorporate T2 images and use them to generate waterfall plots and characterize tumors based on progressive responses, stable responses, etc. is really great and I think the field of GBM tumor immunotherapy needs more papers like this to help move away from just showing survival data. Hopefully, papers like this can help make these types of

techniques a standard of practice pre-clinically.

Reviewer #3 (expertise in Toll-like receptors, drug development):

The authors have addressed that the new work sheds light on a new mechanism of action supported by the phenotyping of the tumor microenvironment.

Dear reviewers,

We thank you for your helpful criticism and comments on our manuscript entitled: “**T cell-independent eradication of experimental glioma by intravenous TLR7/8-agonist-loaded nanoparticles**” by Turco et al. (*Nature Communications NCOMMS-22-18014-T*). During this 2nd revision, we performed additional analysis and generated data to address the points raised by the reviewers. Please find attached our detailed point to point response to the reviewer questions (depicted in gray):

The main criticism that reviewer 2 raised was that: “the CDNP-R848 is not superior to free R848..”

We apologize that we might have caused some confusion based on the way we presented the data. However, we would like to clarify our results and actually feel that our data which is in line with the published literature is sufficiently strong to show that CDNP-R848 is indeed superior to R848.

Regarding group sizes we performed as requested in the first round of revisions a head to head comparison of CDNP-R848 (n=8 mice), R848 (n=9 mice), CDNP and Pbs. This showed that tumor regression occurred in 7/8 CDNP-R848 animals (87.5% of mice; vs 12.5% progressive disease), whereas R848 treatment led to tumor regression only in 1/9 mice (11% regression rate; 55% progressive disease and 33% stable disease). There were also no partial responses but only stable disease in the R848 group. This is in stark contrast to CDNP-R848 which showed partial response in 5/8 mice (62.5%).

We have assembled all data to address this important point in Reviewer Figure 1 including additional analysis of the tumor volume changes that occur upon treatment comparing the baseline (week 2) with the 1st post treatment timepoint (week 3) and the final time point (week 4). This showed highly significant differences for both groups in week 4 (**Reviewer Fig. 1B**). The validity of the experimental design and statistical testing were also confirmed by the head biostatistician of DKFZ (Prof. Holland-Letz). For full transparency we are also adding all MRI images of week 4 (**Reviewer Fig. 1F**) which also show the clear difference of the two treatment formulations. We have added a representative R848 time course to Figure 2b.

F

G

Reviewer Figure 1: C57Bl/6J mice bearing intracranial Gli261 tumors were treated intravenously with 100 μ l CDNP-R848 or R848 on day 14, day 17 and day 20 after baseline MRI and randomization on day 12. Mice were monitored by serial MR imaging on day 12, 19 and day 25. **A** Tumor growth curves (spider plots) and **B** bar graphs showing tumor volumes in the CDNP-R848 and R848 group. n=8 mice for CDNP-R848, n=9 for R848. **C** Response calculation of the two groups based on standardized MRI criteria¹. **D** Tumor volume changes (%) between baseline (week 2), after completion of the treatment cycle (week 3) and the last observation timepoint (week 4). **E** Waterfall plot showing tumor response towards CDNP-R848 and R848 treated mice. **F** T2w MR images of all CDNP-R848 and **G** R848 animals in week 4. Data is shown as mean \pm SEM. Statistical significance was determined by two-way ANOVA with Tukey's test for **B** and **D**. PD: partial response. SD: stable disease. PD: progressive disease.

We have shifted Figure 2c to the supplement as Suppl. Figure 3d,e which assembles all CDNP-R848 experiments that were performed in the study (n=65 CDNP-R848 vs 33 CDNP mice, 6 independent experiments). Here, we find partial or complete response (PR, CR) for CDNP-R848 in 53.8% mice. This very robust treatment effect is truly remarkable also given our previous work¹ on dual checkpoint inhibition by PD1 / CTLA-4 blocking antibodies which only achieved PR/CR in 18.8% of mice (n=212 mice) using the same treatment paradigm, imaging protocol and response criteria. As this cumulative data seemed misleading when compared to the single head to head comparison, we have shifted it to the supplement and now show the head to head comparison instead (Reviewer Figure 1a-c instead; new Figure 2c-f).

The extensive CDNP-R848 data however shows that similar to the head to head comparison of CDNP-R848 and R848 shown in Reviewer Figure 1, only CDNP-R848 achieved CR or PR in a significant proportion of glioma bearing mice in a large cohort of independent experiments.

Furthermore, our findings are well in line with previously published data by Rodell et al² that showed the superiority of CDNP-R848 vs R848 in two additional independent tumor models (MC38 colorectal cancer model and B16 melanoma model, **Reviewer Figure 2**).

Reviewer Figure 2: (Figure reproduced from Rodell et al. ²): Therapeutic efficacy. **a–c**, Efficacy of repeated dosing regimen. **a**, Tumour area at day eight following the start of treatment. Data are expressed as mean \pm s.e.m.; N = 12 tumors; **P = 0.0017, ****P < 0.0001 (Dunn's multiple comparison) relative to vehicle control. **b**, Survival following start of treatment. **P = 0.005 (log-rank test, two-sided) relative to vehicle controls; N = 6 animals. **c**, Macroscopic images of tumors at day eight following initiation of treatment, representative of N = 6 mice per group. **d**, Individual tumour growth curves for mice treated with a single dose of R848 or CDNP-R848. **e**, Change in individual tumour area at day eight following treatment with a single dose of CDNP, CDNP-R848, aPD-1 or the combination therapy. All studies were executed in C57BL/6 mice, and treatment was initiated when tumors reached an area of 25 mm². Figure reproduced from Rodell et al.²

Also, Grauer et al³ have treated Gli261 gliomas with R848 reporting an extension of survival over PBS vehicle control to 35.4 days, whereas we have seen a median survival of CDNP-R848 of 64 days compared to 28 days in CDNP vehicle control mice using the same mouse model and a comparable experimental setup.

Altogether, we are inclined to conclude that the superiority of CDNP-R848 over R848 is sufficiently shown by these different datasets.

Nevertheless, if you think differently, we can perform additional survival experiments. However, these will take considerable time (about 3 to 4 months) and resources. We would also have to discuss performing an additional survival experiment with our

veterinarian given the strict three Rs (3Rs) guidelines that are in place at our institution and the multiple lines of evidence that indicate the superiority of CDNP-R848.

We had shown aggregated data from multiple experiments to show the robustness of the treatment effect. We have exchanged the data in Figure 2 to include the head to head comparison and have shifted the pooled data of CDNP-R848 vs CDNP which were all derived from independent experiments as drug vs vehicle control to the supplement (Suppl. Figure 3c,d).

Reviewer#1:

The results of this manuscript are of potential interest, but fundamental rationale for the observed effect of CDNPs, rather than physiological and *in vivo* response, are required: The tridimensional structure of CDNPs is not straightforward, and the incorporation of R848 is not clear. Also, the role of L-lysine must be explained. Being of particular relevance the findings that nanoparticles improved the readouts, polymer-base or micellar-base structure of the nanocarrier must be elucidated.

We appreciate your interest in the nanoparticle structure. However, these are not open questions and the rationale behind the material design and expected structure have been discussed in detail in the prior publication in which CDNP was developed (Rodell et al. ², 2018). Duplication of this information is not warranted as appropriate citation of the work has already been provided.

Regarding the rationale for material design, it was stated in the introduction of Rodell et al, 2018 ²: "we sought to capitalize on the use of β -cyclodextrin (CD) as a supramolecular drug reservoir. CD has an extensive history in industrial and pharmaceutical applications and an established safety profile; importantly, cyclodextrins are able to form water-soluble inclusion complexes with many poorly soluble drugs, enabling drug solubilization by hydrophilic modified cyclodextrins as well as affinity-based drug delivery when formulated into nanoparticles, surface coatings or bulk materials. We therefore hypothesized that covalent crosslinking of CD would enable formation of cyclodextrin nanoparticles (CDNPs) with macrophage affinity and high drug-loading capacity."

Regarding the use of lysine, this is used as a diamine crosslinker between succinyl groups of the cyclodextrin. These materials were chosen in part because the "base components (for example, L-lysine and cyclodextrin) are recognized by the US Food and Drug Administration as safe for medical use." Crosslinking of the short hydrophilic cyclodextrins by lysine produces roughly spherical nanogels that "yielded optimal properties for systemic delivery (for example, hydrodynamic radius and zeta potential) resulting in $4.1 \pm 1.2\%$ of the injected dose being delivered to a solitary tumour, as compared to a modest 0.7% median for conventional nanoparticle preparations"

Proinflammatory effects described in Figure 1 lack proper controls of the free R848, not encapsulated. One is missing the direct effect of the active molecules.

For *in vitro* cell culture experiments we indeed compared CDNP-R848 to CDNP vehicle control as this seemed the most relevant aspect. Rodell et al. ² have extensively tested R848 vs CDNP-R848 and found that there is some superiority of CDNP-R848 over R848 *in vitro* but the full strength of the formulation is demonstrated mainly *in vivo*², likely because no specific targeting nor uptake competition exists *in vitro*. Also, it has

been shown that R848 works well *in vitro*, penetrates cell membranes and is active *in vitro* to induce its downstream signaling pathways⁴.

The targeting capacity of CDNP-R848 to glioma-associated macrophages and microglia is claimed, but the mechanistic justification is unclear. Likewise, crossing BBB by 20 nm structures is not trivial, and the transport mechanism is absent from the discussion.

We have performed an in depth biodistribution analysis and find indeed that CDNP-R848 preferentially accumulates in glioma-associated macrophages. Microglia are also targeted but to a lesser degree compared to monocytic derived myeloid cells (4.7% vs 25.7%, $p < 0.05$). We have previously performed biodistribution analysis by two photon microscopy using iron oxide NP (mean size of 30nm) that showed intravascular uptake by blood monocytes that become recruited to the TME as well as passive leaking of NP through the disrupted BBB⁵. This is likely similar for CDNP-R848 which has a similar size as the iron oxide NP used in the previous study. We have added an additional paragraph on the targeting mechanism to the revised discussion to address this issue.

Reviewer#2:

Overall, I think this is certainly an interesting paper that was a pleasure to read
Here are some points that I would like to raise:

- In response to many of the comments where the authors were asked to compare CDNP-R848 with R848, the group did two experiments, they showed an unremarkable spider plot and one bar plot showing tumor growth differences at week 7. Based on the data, I am not sure if this is an effect of CDNP-R848 or just an increased accumulation of R848 in the tumor. A telling experiment might be to inject R848 directly into the brain to see if the effect is purely a result of targeting or if there is an added advantage to using this complex.

We have addressed this important point regarding the superiority of CDNP-R848 over R848 in our answer to the Editor's comment (see above).

Concerning the outlined experiment of intratumoral R848 administration Grauer et al have reported on intratumoral R848 injections in the GI261 model previously (doi: 10.4049/jimmunol.181.10.6720, Fig. 3) and found a median survival of 35 days in the GI261 model. As outlined above, our median survival was 65 days for CDNP-R848 with comparable control group survival times in these experiments of 28 days for CDNP vehicle and 27 days for Pbs, respectively. Moreover, we believe the strong translational advance of our study is indeed the intravenous application of CDNP-R848 in contrast to intratumoral injections into the brain which have only little clinical relevance.

- I completely agree with Reviewer 1, Comment 5 about the use of Abs to deplete the immune cells in the periphery. The authors mention having data in supplementary material that I can not see but can not comment on but I find it really surprising that this is what they found to be the case. If this were truly as clean as the authors are suggesting. It would have instead been more interesting to see if these results would hold up in a RAG KO mouse background which is what I have seen more often in these types of brain cancer models. Since the authors claim that their novelty is that their mechanism is T cell and NK cell-independent, it would be important to do the experiments in as clean a system as possible.

We agree that using the RAG2 KO model would be another possibility to show the independence of the treatment effect on adaptive immune cells. This however would require a full set of new large-scale independent experiments in ~ 60 mice (RAG KO vs WT littermates treated during R848, CDNP-R848 and CDNP treatment) which goes beyond the scope of this second round of revisions. As presented in the Suppl. Figure 5e-g we have thoroughly confirmed that the antibody depletion has worked well as assessed by flow cytometry of the tumor bearing hemisphere or by immunohistochemistry. Moreover, we confirmed successful target cell depletion in the peripheral blood before administration of CDNP-R848 to make sure that immune cell depletion had occurred before the start of therapy. Additionally, we monitored successful target cell depletion during the experiment in peripheral blood and within the tumor microenvironment at the end of the experiment. Also, we have successfully utilized the same antibodies using the same depletion protocol in previous work¹ which showed that CD4 depletion potentially abrogated the treatment effect of checkpoint inhibitor blockade in the GI261 glioma model (Aslan et al., 2020, Figure 3h). Together with additional published articles⁶⁻⁸ that use AB depletions successfully (and as we do in the clinical practice with e.g. anti CD20 AB for B-cell depletion in autoimmunity), this indicates that the antibody depletion is a reliable method and that the mode of action of CDNP-R848 is indeed distinct from other established immunotherapies.

Reviewer Figure 3/ Main Figure 4: CDNP-R848 treatment effect is independent of CD8 T cells, and NK cells CD8⁺ T cells, CD4⁺ T cells or NK1.1⁺ were depleted prior to and during CDNP-R848 therapy using monoclonal depletion antibodies. Tumor volume (% between d13 and d26) and response assessment of CD8⁺-depleted or isotype treated mice (CDNP-R848 + CD8 isotype n= 5 vs. CDNP-R848 + α-CD8 depleted, n = 6 animals): Tumor growth of the different treatment and depletion groups. All data are represented as mean ± SEM. Statistical significance was determined by two-tailed Student's test.

Reviewer Figure 4 (Suppl. Fig. 5): Depletion of CD4, CD8 and NK cells does not abrogate CDNP-R848 treatment efficacy **a**: Gating scheme for validation of immune cell depletion experiments. **b-d**: Tumor volumes in CDNP-R848 treated mice that had received α-CD8, α-CD4 or α-NK1.1 to deplete CD8 T cells, CD4 T cells or NK cells in comparison to isotype treated controls. **e-g**: Depletion was confirmed at d27 in the TME by flow cytometry (**e,g**) or by immunofluorescence microscopy (**f**). All data are represented as individual values and the mean ± SEM. Statistical significance was determined by two-tailed student's test.

- Based on the data, is there any chance that the results are due to R848 accumulation in a small compartment leading to immune-independent effects and purely a result of concentration-dependent toxicity?

In the immunohistochemical analysis of the glioma after CDNP-R848 administration we found good targeting of the entire tumor area as shown by immunohistochemistry but no uptake of the glioma cells (no NP-labeled glioma cells). Also, our *in vitro* analysis show that CDNP-R848 is not cytotoxic by itself and does not act as a chemotherapeutic drug that kills proliferating tumor cells. Again, this is in line with the previous report of CDNP-R848 (SFig. 13, Rodell et al)²

Reviewer Figure 5: A MRI and respective histological image shows NP accumulation in the TME of GI261 glioma but not in the adjacent healthy brain. There is no intracellular uptake of NP by GI261 glioma cells. Scale bar is 1mm on MRI and 100μm in micrograph.

- One of the largest limitations aside from those comments already addressed is that this is a study done in GL261 tumors. I find it suspicious that the group would not publish at the very least with CT2As which makes me wonder if their effects are model specific. The Weissleder used MC38s in their experiments. The fact that the authors are using GL261s makes me wonder if there is a PD-1 antagonism play that would not hold up in true GBM tumors.

The superior effect of CDNP-R848 over R848 has been investigated in GI261 glioma and reported previously for the MC38 colon cancer and B16 melanoma model². Of course, additional brain tumor models could be employed to validate our results in additional mouse models. One caveat is, that testing CDNP-R848 requires immuno-competent glioma models and our lab does not work with CT2A. Other immuno-competent mouse brain tumor models our lab has recently developed include neural stem cell-specific Pten/p53 double-knockout models⁹. These however have very different growth dynamics and a median survival over >60 days. Therefore, treatment paradigms, dosing and treatment monitoring would need additional rounds of experiments that go beyond the scope of this revision and should be addressed in upcoming studies.

- I think the methods here to evaluate tumor responses are really remarkable and commendable. The fact that they were able to incorporate T2 images and use them to generate waterfall plots and characterize tumors based on progressive responses, stable responses, etc. is really great and I think the field of GBM tumor immunotherapy needs more papers like this to help move away from just showing survival data. Hopefully, papers like this can help make these types of techniques a standard of practice pre-clinically.

Thank you for the positive comment. We agree that MRI has large advantages over other modalities used in the field, such as bioluminescence imaging, computed

tomography or survival studies. Tumor response assessment should be performed by the best and most reliable modality and that is MRI. Though this needs dedicated infrastructure, MRI is the clinical gold standard and allows the conduction of a clinical trial-like study and reporting system.

Reviewer#3:

The authors have addressed that the new work sheds light on a new mechanism of action supported by the phenotyping of the tumor microenvironment.

Thanks for the positive assessment of our work.

References:

1. Aslan, K. *et al.* Heterogeneity of response to immune checkpoint blockade in hypermutated experimental gliomas. *Nature Communications* 1–14 (2020) doi:10.1038/s41467-020-14642-0.
2. Rodell, C. B. *et al.* TLR7/8-agonist-loaded nanoparticles promote the polarization of tumour-associated macrophages to enhance cancer immunotherapy. *Nature Biomedical Engineering* 1–15 (2018) doi:10.1038/s41551-018-0236-8.
3. Grauer, O. M. *et al.* TLR Ligands in the Local Treatment of Established Intracerebral Murine Gliomas. *J Immunol* 181, 6720–6729 (2008).
4. Bourquin, C. *et al.* Systemic Cancer Therapy with a Small Molecule Agonist of Toll-like Receptor 7 Can Be Improved by Circumventing TLR Tolerance. *Cancer Res* 71, 5123–5133 (2011).
5. Karimian-Jazi, K. *et al.* Monitoring innate immune cell dynamics in the glioma microenvironment by magnetic resonance imaging and multiphoton microscopy (MR-MPM). *Theranostics* 10, 1873–1883 (2019).
6. Kim, J. *et al.* Memory programming in CD8+ T-cell differentiation is intrinsic and is not determined by CD4 help. *Nat Commun* 6, 7994 (2015).
7. Wu, J. & Waxman, D. J. Metronomic cyclophosphamide eradicates large implanted GL261 gliomas by activating antitumor Cd8+ T-cell responses and immune memory. *Oncoimmunology* 4, e1005521 (2015).
8. Butler, N. S. *et al.* Therapeutic blockade of PD-L1 and LAG-3 rapidly clears established blood-stage Plasmodium infection. *Nat Immunol* 13, 188–195 (2012).
9. Costa, B. *et al.* A Set of Cell Lines Derived from a Genetic Murine Glioblastoma Model Recapitulates Molecular and Morphological Characteristics of Human Tumors. *Cancers* 13, 230 (2021).

REVIEWERS' COMMENTS

Reviewer #1 (Remarks to the Author):

The authors properly addressed the previous queries.

Reviewer #2 (Remarks to the Author):

The authors have addressed my comments and questions satisfactorily